# AI-assisted radiologists vs. standard double reading for rib fracture detection on CT images: A real-world clinical study

Li Sun[1☉], Yangyang Fan[1☉], Shan Shi[1], Minghong Sun[1], Yunyao Ma[1], Kuo Zhang[1], Feng Zhang[1], Huan Liu[1], Tong Yu[2], Haibin Tong[1], Xuedong Yang[1]*

1 Department of Radiology, Guang'anmen Hospital, China Academy of Chinese Medical Sciences, Beijing, China, 2 Department of Orthopedic, Guang'anmen Hospital, China Academy of Chinese Medical Sciences, Beijing, China

☉ These authors contributed equally to this work.
* yangxuedong1@163.com

**Data Availability Statement:** All relevant data are within the manuscript and its Supporting Information files.

## Abstract

To evaluate the diagnostic accuracy of artificial intelligence (AI) assisted radiologists and standard double-reading in real-world clinical settings for rib fractures (RFs) detection on CT images. This study included 243 consecutive chest trauma patients (mean age, 58.1 years; female, 166) with rib CT scans. All CT scans were interpreted by two radiologists. The CT images were re-evaluated by primary readers with AI assistance in a blinded manner. Reference standards were established by two musculoskeletal radiologists. The re-evaluation results were then compared with those from the initial double-reading. The primary analysis focused on demonstrate superiority of AI-assisted sensitivity and the noninferiority of specificity at patient level, compared to standard double-reading. Secondary endpoints were at the rib and lesion levels. Stand-alone AI performance was also assessed. The influence of patient characteristics, report time, and RF features on the performance of AI and radiologists was investigated. At patient level, AI-assisted radiologists significantly improved sensitivity by 25.0% (95% CI: 10.5, 39.5; P < 0.001 for superiority), compared to double-reading, from 69.2% to 94.2%. And, the specificity of AI-assisted diagnosis (100%) was noninferior to double-reading (98.2%) with a difference of 1.8% (95% CI: -3.8, 7.4; P = 0.999 for noninferiority). The diagnostic accuracy of both radiologists and AI was influenced by patient gender, rib number, fracture location, and fracture type. Radiologist performance was affected by report time, whereas AI's diagnostic accuracy was influenced by patient age and the side of the rib involved. AI-assisted additional-reader workflow might be a feasible strategy to instead of traditional double-reading, potentially offering higher sensitivity and specificity compared to standard double-reading in real-word clinical practice.

## Introduction

Rib fractures (RFs) are commonly encountered in clinical practice. Given that traumatic RFs are associated with higher mortality and morbidity, a quick and accurate diagnosis is of clinical

**Funding:** 1. Scientific and technological innovation project of China Academy of Chinese Medical Sciences (No. CI2021A03312 to L.S.), China Academy of Chinese Medical Sciences 2. Guang'anmen Hospital of China Academy of Traditional Chinese Medicine escort project - key talent cultivation program (No. 9323018 to L.S.), Guang'anmen Hospital, China Academy of Chinese Medical Sciences.

**Competing interests:** The authors have declared that no competing interests exist.

importance [1]. Multidetector computed tomography (MDCT), owing to its superior in-plane resolution and reduced slice thickness and spacing, has become the preferred modality for radiologic evaluation of trauma patients [2]. MDCT is advantageous for identifying RFs and associated complications, offering higher sensitivity and specificity compared to plain radiography [3]. However, despite the high-quality and multiplanar reformatted images provided by MDCT, missed RFs remain a common issue in daily clinical practice [4].

Double-reading is one of the ways to increase the quality of radiology reports. This process involves two radiology specialists interpreting the same study, either as peers with similar experience levels or through secondary reading by a higher level of sub-specialization [5]. Trauma CT is one of the areas where double-reading seems to be important [6]. Nevertheless, practical constraints such as radiologist availability and workload often limit the widespread adoption of this practice. Furthermore, the potential for systematic errors due to fatigue and high work-load remains an inherent challenge in human diagnosis [7, 8].

Deep learning shows potential in reducing the workload of image interpretation and minimizing diagnostic oversight [9–12]. Studies have demonstrated that AI can achieve diagnostic efficiency comparable to that of experienced radiologists in controlled settings, with shorter reading time [13–18]. However, the effectiveness of AI in real-world clinical scenarios, especially in conjunction with double-reading, and its performance consistency across diverse cases require further investigation.

This study aimed to evaluate the diagnostic accuracy of AI-assisted radiologists (radiologist plus AI) and standard double reading by dual-radiologists in real-world clinical settings for RF detection on CT scans. The influence of patient characteristics, report timing, and RF features on performance of both AI and radiologists will also be assessed, to provide a nuanced understanding of the AI's potential in augmenting radiological diagnostics.

## Materials and methods

The study was approved by our Ethics Committee (No. 2024-058-KY; 03/04/2024), with informed consent waived in line with guidelines.

### 1. Dataset collection and access

Our dataset included rib CT scans of 243 consecutive chest trauma patients (mean age, 58.1 years; female, 166) admitted to our emergency department from January 1, 2019, to December 31, 2020, with 188 patients (mean age, 61.1 years; female, 127) diagnosed with RFs and 55 patients (mean age, 46.6 years; female, 39) serving as non-RF controls (Table 1). Intrathoracic injuries associated with RFs included pneumothorax (5.3%, 10 of 188), pulmonary contusions

**Table 1. Characteristics of patients and RFs.**

| Index | RF | Non-RF | Total | P value |
|---|---|---|---|---|
| Number | 188 (77.4) | 55 (22.6) | 243 | - |
| Age (years) | 61.1±15.0* | 46.6±17.8* | | <0.001 |
| Sex | | | | |
| Male | 61 (79.2) | 16 (22.8) | 77 | 0.638 |
| Female | 127 (76.5) | 39 (23.5) | 166 | |
| Number of ribs | 721 | 5083 | 5804 | - |

Note—Unless otherwise indicated, the data represent the counts of observations, and data in parentheses are percentages.

* Data are means ± SDs.

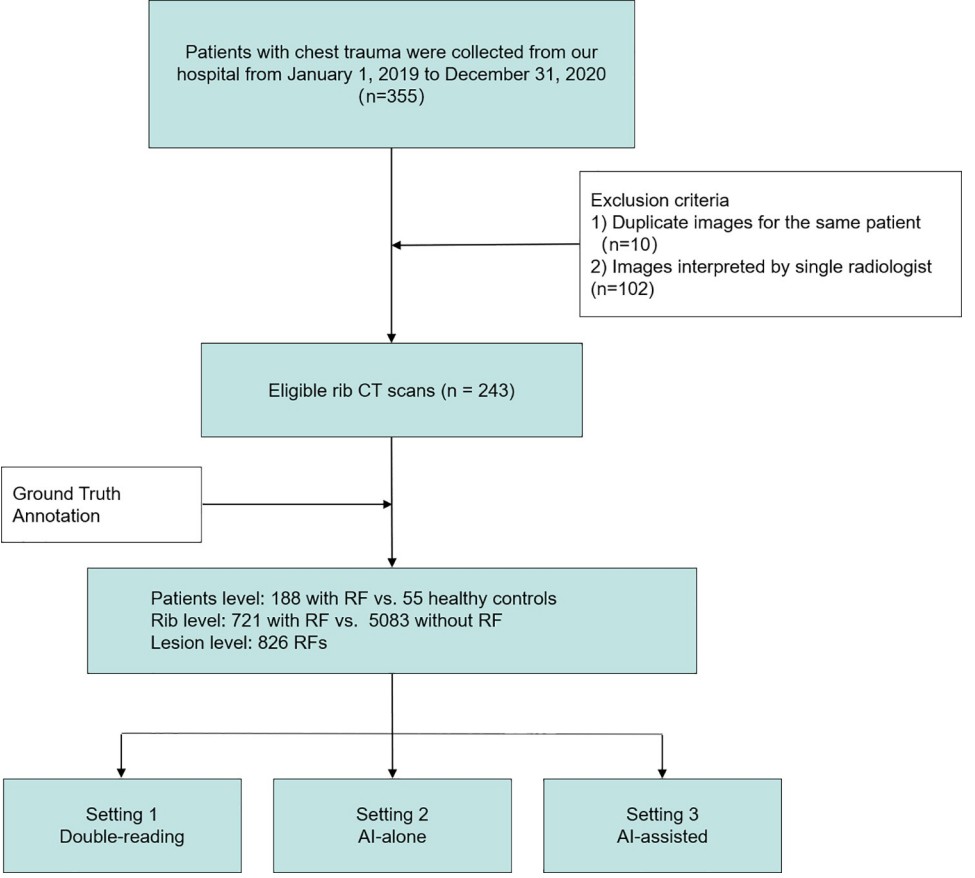

**Fig 1. Flowchart of overall study process.**

(2.1%, 4 of 188), and hemothorax (1.1%, 2 of 188). Single-reader interpretations and duplicate images were excluded (Fig 1). The patient data was accessed for research purposes between 4th April to 31st May 2024. All images were reviewed through the picture archiving and communication system (PACS) (3.0.30500.213; Hinacom Software and Technology, Beijing, China) on a Windows 7 system (Microsoft Corporation, Redmond, Washington, United States) using professional radiographic monitors (JUSHA-M52C, JUSHA-M53AA, JUSHA-M33B and JUSHA-M33D; Nanjing Jusha Display Technology, Nanjing, China). Authors had access to information that could identify individual participants during and after data collection.

Scans were performed on two Siemens CT scanners (Emotion 16 and SOMATOM Definition Flash) with a 120 kV tube voltage and automated dose modulation, with reconstruction layer thicknesses of 1.5 mm for Emotion 16 and 0.75 mm for SOMATOM Definition Flash, using a bone algorithm.

## 2. AI system

The AI analysis utilized a commercially available AI system (uAI-BoneCare; version 20220730SP1; Shanghai United Imaging Intelligence, Co., Ltd.), which is a medical device intended to detect RF on chest CT scans. The pipeline of this AI system consists of five steps: rib segmentation (using modified V-Net model), vertebrae detection (using VRB-Net), rib labelling, RF detection (using the DL-based detection model VRB-Net), and RF classification

(using DL-based classification model, BasicNet) [17]. The data set used in the present study was not used for training, validation, or testing of this AI software.

## 3. Reference standard and lesion definition

Two musculoskeletal radiologists, with over a decade of experience, reviewed all CT examinations in consensus to establish the "ground truth", and the presence or absence of fractures was determined based on the imaging features of RF, clinical history, and findings from previous or follow-up CT scans, in April 2024. Additionally, the number of involved ribs from the 1st to 13th (where the 13th rib corresponds to the lumbar ribs), side (left or right), fracture location, and fracture type were documented. Fracture location was categorized into five anatomic regions: near the junction of the costal cartilage (Nj), anterior (A), lateral (L), posterior (P), and near the vertebral bodies (Nv), as depicted in Fig 2. Each lesion was classified into four main types based on morphology and age: (I) fresh fracture with no or minimal dislocation, (II) fresh fracture with distinct dislocation, (III) healing fracture with periosteal reaction or

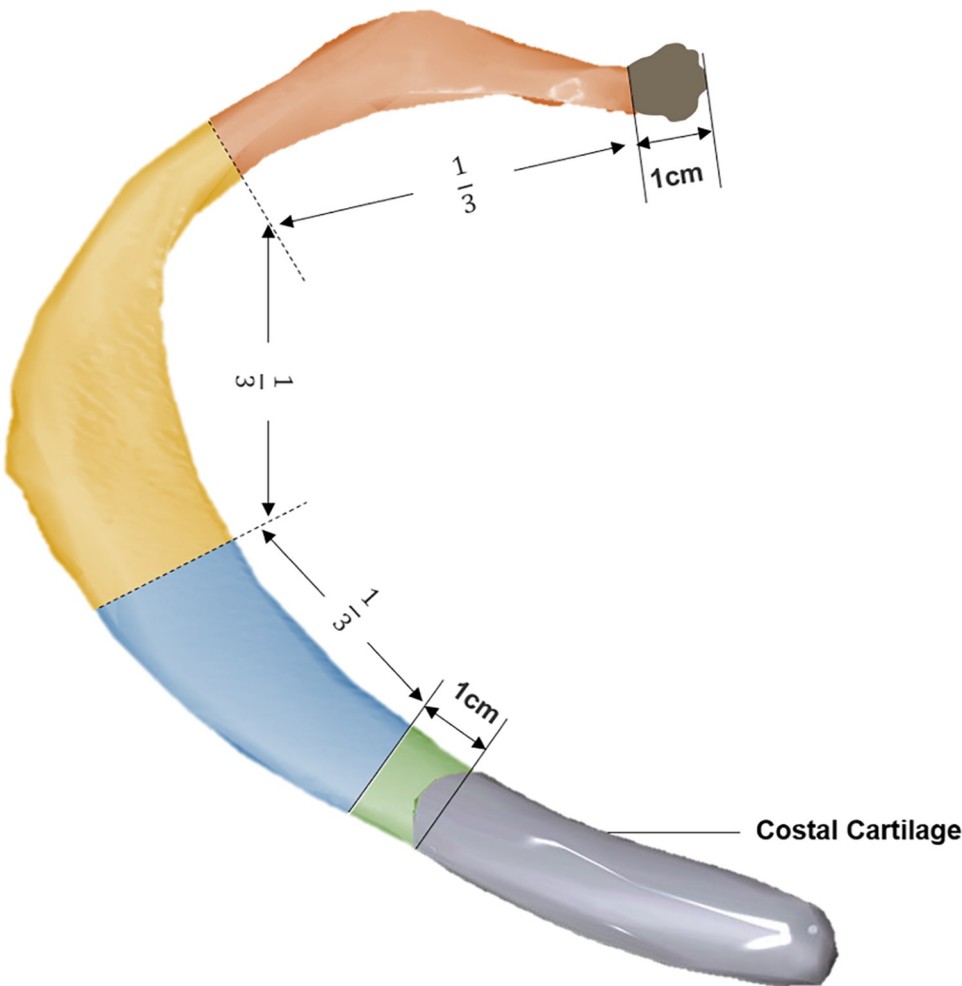

**Fig 2. Schematic diagram of five anatomic regions of rib arc.** Nj: within 1cm of junction of costal cartilage (green). Nv: within 1cm of costovertebral junction. A: anterior 1/3 length between Nj and Nv (blue). L: middle 1/3 length between Nj and Nv (yellow). P: posterior 1/3 length between Nj and Nv (orange). The 11th and 12th ribs don't have Nj. (Nj = near the junction of costal cartilage, A = anterior, L = lateral, P = posterior, Nv = near vertebral bodies).

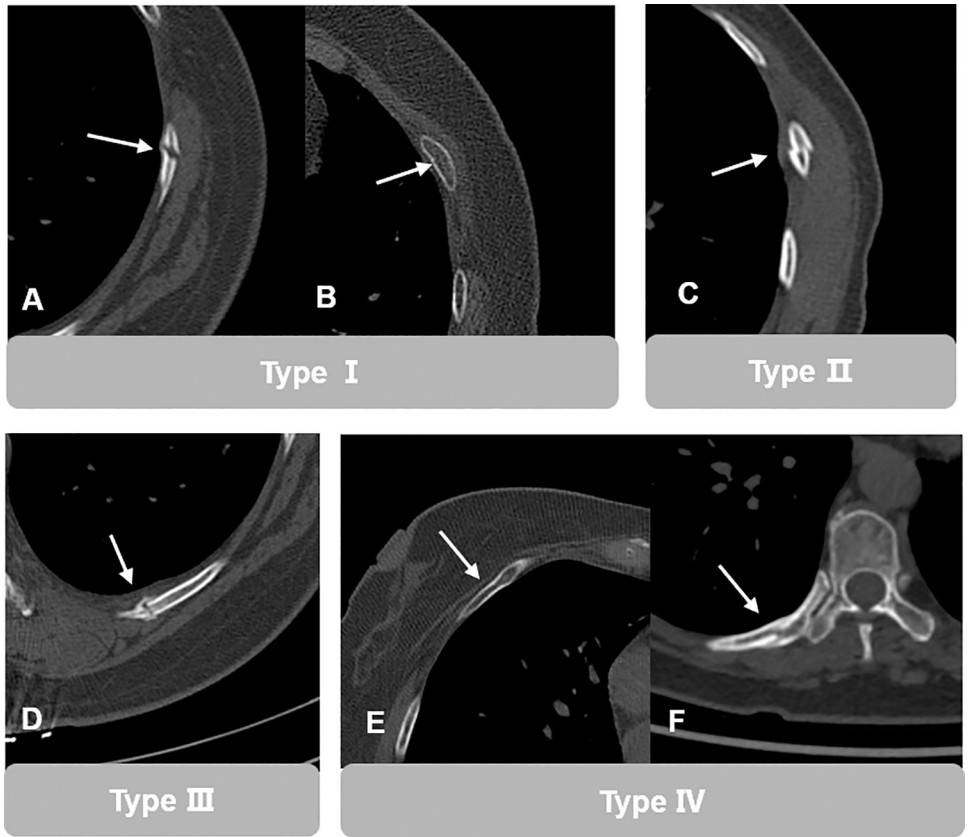

**Fig 3. Demonstration of fracture types.** (I) Fresh fracture with no or minimal dislocation (A and B); (II) fresh fracture with distinct dislocation (C); (III) healing fracture with periosteal reaction or callus formation (D); (IV) old fracture lack of periosteal reaction or callus formation (E and F).

callus formation, (IV) old fracture lacking periosteal reaction or callus formation. These types are illustrated in Fig 3.

## 4. Image interpretation

This study compared the diagnostic efficiency of double-reading by two radiologists, AI-alone, and AI-assisted diagnosis. Double-reading diagnoses were directly extracted from our hospital's PACS from January 1, 2019, to December 31, 2020. AII primary reports were conducted by 2 residents (2nd and 3rd year of residency; female, 1) and 5 attending radiologists (4 to 8 years of experience; mean, 6.6±1.7 years of experience; female, 5), with secondary readings performed by senior radiologists with over 10 years of experience (See S1 Table for details). For AI-assisted diagnosis, to simulate real-world clinical scenarios, from April to May 2024, the CT images of each patient were integrated back into the daily workload of the original junior radiologists who had initially interpreted the cases via PACS based on the original scan timestamps. This time, radiologists read CT images with the assistance of AI. AI-alone diagnoses were drawn from the structured reports automatically formed by the AI system. Diagnostic outcomes were compared against the reference standard (Fig 1).

The diagnostic outcomes from the double-reading, AI-assisted groups, and AI-alone were compared against the reference standard. The comparisons were made at the patient, rib, and lesion levels to categorize the results as true positives, false negatives, true negatives, and false

positives. These categorizations were based on the classification rules outlined in the study by Liu et al. [19].

## 5. Statistical analyses

**Principles of statistical analysis.** Statistical analysis was performed using IBM SPSS Statistics (Version 27.0, IBM Corp., Armonk, N.Y., USA). Continuous variables, age, were reported as mean ± standard deviation (SD). Group comparisons for continuous variables were performed using independent samples t-tests. Categorical variables were summarized using frequencies and percentages. Group comparisons for categorical variables were evaluated using chi-square tests.

The DeLong test was used to compare the areas under the receiver operating characteristic (ROC) curves (AUCs) and provided a statistical measure of the difference. The McNemar test was used to analyze diagnostic performance by comparing sensitivity and specificity.

All statistical tests were two-tailed, and the significance level was set at $p < 0.05$. 95% confidence intervals (CIs) were reported.

**Primary endpoint.** The primary endpoints of this study focused on patient-level sensitivity and specificity. The trial aimed to demonstrate that AI-assisted diagnosis had superior sensitivity and non-inferior specificity compared to double-reading. Superior sensitivity was claimed if the lower bound of the 95% CI for the sensitivity difference was >0. Non-inferiority of specificity was established if the lower bound of the 95% CI for the specificity difference was greater than −10%.

**Secondary endpoint.** Secondary endpoints included statistical analysis of sensitivity and specificity at the rib level, along with AUC comparisons at both the patient and rib levels. Lesion-level assessments included sensitivity and precision comparisons between AI-assisted and double-reading groups. Furthermore, the performance of the AI-alone was evaluated against the double-reading.

## Results

### 1. RF characteristics

A total of 826 RF lesions were identified across 721 ribs, while the remaining 5083 ribs were free from fractures (Table 1). 328 RFs classified as acute, with 245 as Type I and 83 as Type II. Old fractures, constituting 498 RFs, included 123 Type III and 375 Type IV. The spatial distribution of RFs within the rib arcs is detailed in S1 Fig.

### 2. Primary performance metrics

The sensitivity at patient level was estimated at 94.2% in AI-assisted group and 69.2% in double-reading, with an increase estimated at 25.0% (95% CI: 10.5, 39.5; P < 0.001). It indicated that the sensitivity of RF detection by AI-assisted was superior to that of double-reading. Specificity was perfect at 100% for AI-assisted diagnosis versus 98.2% for double-reading, showing a non-significant increase of 1.8% (95% CI: -3.8, 7.4; P = 0.999). It suggested that the specificity of RF detection by AI-assisted diagnosis was not inferior to double-reading. Primary performance metrics analysis was thus successful (Table 2).

### 3. Secondary performance metrics

Rib-level analysis revealed that AI-assisted diagnosis sensitivity significantly higher than the double-reading (P < 0.001), with an increase of 23.2% (95% CI: 0, 46.4), and specificity with a non-significant increase of 0.1% (95% CI: -0.3, 0.5; P = 0.998). The AI-assisted group

**Table 2. Performance comparison of AI-assisted and AI-alone diagnosis versus double-reading at patient, rib, and lesion levels.**

| Index | Double-reading | AI-alone | Difference † | p-value* | AI-assisted | Difference † | p-value* |
|---|---|---|---|---|---|---|---|
| **Patient level** | | | | | | | |
| Sensitivity | 69.2 (130/188) | 95.7 (178/186) | 26.5 [12.4–40.7] | <0.001 | 94.2 (177/188) | 25 [10.5–39.5] | <0.001 |
| Specificity | 98.2 (54/55) | 94.7 (54/57) | -3.5 [-11.4 to 4.4] | 0.999 | 100 (55/55) | 1.8 [-3.8 to 7.4] | 0.999 |
| AUC | 0.837 | 0.97 | 0.1 [0.1–0.2] | <0.001 | 0.971 | 0.1 [0.1–0.2] | <0.001 |
| **Rib level** | | | | | | | |
| Sensitivity | 49.8 (359/721) | 70.9 (510/719) | 21.1 [-2.2 to 44.4] | <0.001 | 73.0 (526/721) | 23.2 [0–46.4] | <0.001 |
| Specificity | 99.9 (5079/5083) | 99.6 (5066/5085) | -0.3 [-1.2 to 0.6] | 0.998 | 100.0 (5082/5083) | 0.1 [-0.3 to 0.5] | 0.998 |
| AUC | 0.749 | 0.853 | 0.1 [0–0.2] | 0.20 | 0.865 | 0.1 [0–0.2] | 0.25 |
| **Lesion level** | | | | | | | |
| Sensitivity | 51.0 (421/826) | 70.6 (583/826) | 19.6 [11.7–27.5] | <0.001 | 72.8 (601/826) | 21.8 [13.8–29.8] | <0.001 |
| precision | 99.1 (421/425) | 96.5 (583/604) | -2.6 [-5.3 to 0.1] | 0.880 | 99.8 (601/602) | 0.7 [0–1.4] | 0.999 |

Note—Unless otherwise indicated, data are percentages, and numbers in parentheses denote the respective numerators and denominators.

* The p-value for significance analysis using double-reading as a comparison.

† The difference indicates the values of the AI-assisted group/AI-alone group–the values of the double-reading group. The data enclosed in brackets correspond to the 95% confidence intervals.

demonstrated a significantly higher AUC compared to the double-reading group at the patient level (P < 0.001) (Table 2).

For lesion-level analysis, 25 lesions were misdiagnosed as fractures by radiologists or AI, resulting in false positives. The sensitivity of AI-assisted diagnosis was significantly higher than that of double-reading (P < 0.001). No significant difference was found between precision of two groups (P = 0.999) (Table 2).

AI-alone demonstrated significantly higher sensitivity at patient, rib, and lesion levels and significant higher AUC at patient level than double-reading (all P < 0.001). AUC at rib level (P = 0.20), specificity at patient level (P = 0.999) and rib level (P = 0.998), and precision at lesion level (P = 0.880) was similar between AI-alone and double-reading (Table 2).

## 4. Influences on diagnostic accuracy at lesion level: Patient characteristics, report time, and RF features

Both radiologists and AI demonstrated a higher diagnostic accuracy for male patients than for females (p < 0.001). Age did not impact the radiologists' diagnostic accuracy (p = 0.802); however, AI showed a decrease in accuracy with increasing patient age (p < 0.001) (Fig 4 and Table 3).

Radiologists' diagnostic accuracy varied significantly by reporting time (p < 0.001). Notably, AI demonstrated consistent diagnostic performance throughout all reporting times, with no significant variation (p = 0.421) (Fig 4 and Table 3).

Rib number significantly influenced the diagnostic accuracy of both radiologists (p = 0.001) and AI (p = 0.002). Fracture location and type also significantly affected the diagnostic accuracy for both groups (p < 0.001). Radiologists showed no significant difference in diagnostic accuracy for left versus right RFs (p = 0.197). In contrast, AI exhibited a significant higher accuracy for left-sided than for right-sided fractures (p = 0.029) (Fig 4 and Table 4).

## Discussion

Previous studies provide evidence of AI's potential to enhance diagnostic ability of both radiologists and non-radiologists [11, 20]. In time-consuming and labor-intensive works, such as

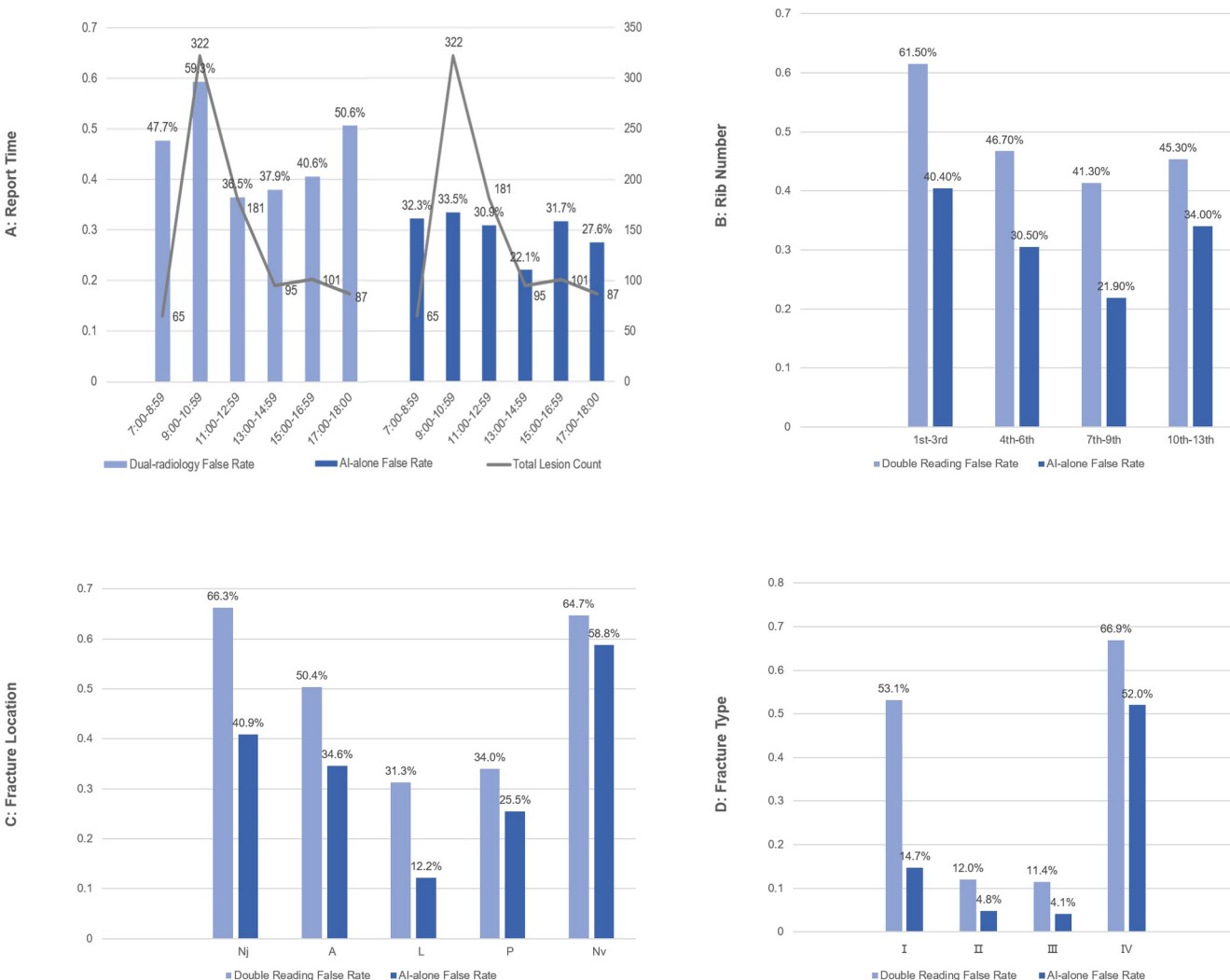

**Fig 4. Bar graph of the impact of report time, rib number, fracture location, and fracture type on the diagnostic performance of double-reading and AI-alone. A:** Radiologists showed lowest accuracy in the early morning, aligning with the highest lesion volume, suggesting a workload effect, and A gradual afternoon decline in accuracy was observed. AI demonstrated consistent diagnostic performance throughout all reporting times. **B:** Rib number significantly influenced the diagnostic accuracy of both radiologists and AI, with the upper ribs (1st-3rd) being more challenging to diagnose correctly. **C:** Fracture location significantly affected the diagnostic accuracy, with least accurate diagnoses in Nv and Nj for both AI and radiologists. **D:** Fracture type significantly affected the diagnostic accuracy, with least accurate diagnoses for fractures with subtle morphological changes (such as Type I and IV). Notably, AI outperformed dual radiologists in diagnosing type I fractures (85.3% vs. 46.9%). (Nj = near the junction of costal cartilage, A = anterior, L = lateral, P = posterior, Nv = near vertebral bodies).

RFs detection, AI-supported diagnosis is presumed to resulted in a similar accuracy, with shorter reading time [11–18, 21], however, evidence of the safe implementation of AI in real clinical practice is limited. The stand-alone performance of our AI system (AUC, 0.97; sensitivity, 95.7%) stands in concert with the high benchmarks set by earlier studies (AUC, > 0.90; sensitivity, > 90% for most studies) [11–14, 16, 18]. Notably, our study did not pre-screen images to eliminate factors that would reducing diagnostic accuracy, such as significant artifacts, osseous neoplasms, congenital rib abnormalities, or a history of rib surgery [13–18]. Despite this more challenging and unfiltered dataset, our AI software demonstrated comparable diagnostic stability, indicating its robustness and adaptability to the complexities of clinical practice.

**Table 3. The impact of patient characteristics and report timing on performance of AI and radiologists.**

| Index | Double-reading | | | | AI-alone | | | |
|---|---|---|---|---|---|---|---|---|
| | True | False | Total | P value | True | False | Total | P value |
| **Sex** | | | | | | | | |
| Male | 190 (67.1) | 93 (32.9) | 283 | <0.001 | 231 (81.6) | 52 (18.4) | 283 | <0.001 |
| Female | 252 (44.4) | 316 (55.6) | 568 | | 358 (63.0) | 210 (37.0) | 568 | |
| **Age** | 65.47±14.28* | 65.21±15.77* | | 0.802 | 63.54±14.95* | 69.40±14.36* | | <0.001 |
| **Report Time** | | | | | | | | |
| 7:00–8:59 | 34 (52.3) | 31 (47.7) | 65 | <0.001 | 44 (67.7) | 21 (32.3) | 65 | 0.412 |
| 9:00–10:59 | 131 (40.7) | 191 (59.3) | 322 | | 214 (66.5) | 108 (33.5) | 322 | |
| 11:00–12:59 | 115 (63.5) | 66 (36.5) | 181 | | 125 (69.1) | 56 (30.9) | 181 | |
| 13:00–14:59 | 59 (62.1) | 36 (37.9) | 95 | | 74 (77.9) | 21 (22.1) | 95 | |
| 15:00–16:59 | 60 (59.4) | 41 (40.6) | 101 | | 69 (68.3) | 32 (31.7) | 101 | |
| 17:00–18:00 | 43 (49.4) | 44 (50.6) | 87 | | 63 (72.4) | 24 (27.6) | 87 | |

Note—Unless otherwise indicated, the data represent the counts of observations, and data in parentheses are percentages.

* Data are means ± SDs.

The AI-assisted performance of junior radiologists in clinical scenarios was impressive, with an AUC of 0.97 and a sensitivity of 95.7% at the patient level, which is in keeping with the high standards established by prior researches [12–15]. Importantly, this represents a significant 25.0% (95% CI: 10.5, 39.5) increase in sensitivity over the double-reading method

**Table 4. The impact of features of RF on performance of AI and radiologists.**

| Index | Double-reading | | | | AI-alone | | | |
|---|---|---|---|---|---|---|---|---|
| | True | False | Total | P value | True | False | Total | P value |
| **Rib number** | | | | | | | | |
| 1st-3rd | 62 (38.5) | 99 (61.5) | 161 | 0.001 | 96 (59.6) | 65 (40.4) | 161 | 0.002 |
| 4th-6th | 204 (53.3) | 179 (46.7) | 383 | | 266 (69.5) | 117 (30.5) | 383 | |
| 7th-9th | 118 (58.7) | 83 (41.3) | 201 | | 157 (78.1) | 44 (21.9) | 201 | |
| 10th-13th | 58 (54.7) | 48 (45.3) | 106 | | 70 (66.0) | 36 (34.0) | 106 | |
| **Side** | | | | | | | | |
| Left | 240 (54.1) | 204 (45.9) | 444 | 0.197 | 322 (72.5) | 122 (27.5) | 444 | 0.029 |
| Right | 202 (49.6) | 205 (50.4) | 407 | | 267 (65.6) | 140 (34.4) | 407 | |
| **Location** | | | | | | | | |
| Nj | 65 (33.7) | 128 (66.3) | 193 | <0.001 | 114 (59.1) | 79 (40.9) | 193 | <0.001 |
| A | 169 (49.6) | 172 (50.4) | 341 | | 223 (65.4) | 118 (34.6) | 341 | |
| L | 101 (68.7) | 46 (31.3) | 147 | | 129 (87.8) | 18 (12.2) | 147 | |
| P | 101 (66.0) | 52 (34.0) | 153 | | 114 (74.5) | 39 (25.5) | 153 | |
| Nv | 6 (35.3) | 11 (64.7) | 17 | | 7 (41.2) | 10 (58.8) | 17 | |
| **Type** | | | | | | | | |
| I | 115 (46.9) | 130 (53.1) | 245 | <0.001 | 209 (85.3) | 36 (14.7) | 245 | <0.001 |
| II | 73 (88.0) | 10 (12.0) | 83 | | 79 (95.2) | 4 (4.8) | 83 | |
| III | 109 (88.6) | 14 (11.4) | 123 | | 118 (95.9) | 5 (4.1) | 123 | |
| IV | 124 (33.1) | 251 (66.9) | 375 | | 180 (48.0) | 195 (52.0) | 375 | |

Note—Unless otherwise indicated, the data represent the counts of observations, and data in parentheses are percentages. Nj = near the junction of costal cartilage, A = anterior, L = lateral, P = posterior, Nv = near vertebral bodies.

supervised by senior radiologists (69.2%; P < 0.001), with non-inferior specificity. The enhanced sensitivity of AI-assisted radiologists was also evident at the rib and lesion levels, highlighting AI's potential in augmenting radiological diagnostics in the context of RF detection in real clinical settings. The findings provide compelling evidence for implementing AI in double-reading to reducing workload in RF evaluation for chest blunt trauma without sacrificing diagnostic accuracy.

The excellent diagnostic capability achieved by AI in conjunction with radiologists may be related to the complementarity of AI and human diagnostic abilities. During the diagnostic process, they may be influenced by different factors. For example, as depicted in the present study, performance of radiologist fluctuated across the day reflecting the cognitive errors due to human fatigue [7, 8]. In contrast, our results demonstrated AI maintained a consistent accuracy rate, not subject to the same fatigue-related limitations. While, older age was one of the factors that decrease AI's diagnostic accuracy. This may be due to the challenge for AI algorithm in distinguishing between age-related bone changes and actual fractures. Radiologists, however, consisted with previous studies, were not influenced by age [22], indicating the role of human expertise in mitigating such biases. Our results also demonstrated that fresh fractures, particularly type I, were more accurately diagnosed by AI, suggesting it can detect subtle lesions that may be missed by the human eye [9–11].

Certain factors can influence the diagnostic accuracy of both AI and radiologists, but their reasons may differ. Radiologists showed lower accuracy for female and upper RFs may be due to the anatomical features, which may potentially affect the visibility of fracture, such as the smaller cross-sectional area and cortical thickness of female than male [23]. While, AI's decreasing in accuracy may be attributed to the relatively insufficient in training dataset. For example, the relatively inferior diagnostic capability in upper rib may be attributed to the lower incidence of fractures in these regions, as are shielded by surrounding structures, which result in the relatively insufficient in training dataset [18]. As our study shows that RFs are approximately normally distributed, with relatively lower incidences of fractures in the upper and lower portion of rib cage, which also gained the relatively lowest diagnostic efficiency the present study. Moreover, challenges in diagnosing fractures in the anterior and posterior junctions of the rib were noted. The anatomical complexities of these areas pose challenges for both human and AI to detect RFs [4, 18, 22], emphasizing the need for caution when integrating AI into clinical practice, especially in junction areas.

This preliminary study had several limitations. First, data were collected from only one center, and only one commercial AI system was analyzed in the present study. The results needed to validate by more data from multiple centers and implementing other commercially available systems. Second, given that the interval between the first and second evaluations, during these 4–5 years, average workload of in our department increased by 20–40%. As a result, AI-assisted radiologists have achieved better diagnostic performance under greater work pressure. Third, since this is a retrospective study, the time spent on diagnosis cannot be calculated. Previous studies have consistently demonstrated the ability of AI to shorten the reading time of radiologist in diagnosing RFs in chest CT [12–15]. With the aid of AI, report time savings ranged from approximately 73.9 seconds to 771 seconds, depending on the study conditions [12, 15]. Meanwhile, the results of this study showed AI's potential to replace one of the radiologists in traditional double reading, which would reduce the workload.

In conclusion, our results indicate AI-assisted additional-reader workflow might be a feasible strategy to reducing workload in RF evaluation in real clinical practice. The present study highlighting the potential of AI to offer consistent diagnostic performance across varying conditions and its superiority in detecting certain types of fractures, while, further efforts are

needed to refine AI algorithms to better accommodate the nuances of patient-specific factors and further enhance its diagnostic capabilities in collaboration with radiologists.

## Supporting information

**S1 Table. Information of radiologists.**
(DOCX)

**S1 Fig. Distribution of the number of RFs (n = 826).** The most common fracture location was the region A (40.2%) followed by Nj (23.4%). The prevalence of RFs approximated a Gaussian distribution, with the apex appearing in the fifth rib. More than 56% of all RFs were between the 4th and 7th ribs. (Nj = near the junction of costal cartilage, A = anterior, L = lateral, P = posterior, Nv = near vertebral bodies).
(TIF)

## Acknowledgments

I am grateful for the guidance and support provided by Jingjing Cui and Yaoling Shen in the field of statistics.

## Author Contributions

**Conceptualization:** Haibin Tong, Xuedong Yang.

**Data curation:** Li Sun, Yangyang Fan, Shan Shi, Minghong Sun.

**Formal analysis:** Li Sun, Yangyang Fan.

**Funding acquisition:** Li Sun.

**Investigation:** Li Sun, Yangyang Fan, Shan Shi, Minghong Sun, Yunyao Ma, Kuo Zhang, Feng Zhang, Huan Liu, Tong Yu.

**Methodology:** Yangyang Fan, Xuedong Yang.

**Project administration:** Yangyang Fan.

**Supervision:** Tong Yu, Haibin Tong, Xuedong Yang.

**Validation:** Yangyang Fan.

**Visualization:** Li Sun, Yangyang Fan.

**Writing – original draft:** Li Sun.

**Writing – review & editing:** Xuedong Yang.

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
