## [Decision Letter · Decision Letter 0]

22 Nov 2024

PONE-D-24-35102AI-Assisted Radiologists vs. Standard Double Reading for Rib Fracture Detection on CT Images: A Real-World Clinical StudyPLOS ONE

Dear Dr. Yang,

Thank you for submitting your manuscript to PLOS ONE. After careful consideration, we feel that it has merit but does not fully meet PLOS ONE’s publication criteria as it currently stands. Therefore, we invite you to submit a revised version of the manuscript that addresses the points raised during the review process.

We look forward to receiving your revised manuscript.

Kind regards,

Priti Chaudhary, M.S.

Academic Editor

PLOS ONE

Journal Requirements:

2. Thank you for stating the following financial disclosure: “Funding: 1. Scientific and technological innovation project of China Academy of Chinese Medical Sciences (No. CI2021A03312)

2. Guanganmen Hospital of China Academy of Traditional Chinese Medicine escort project - key talent cultivation program (9323018)”

3. We note that your Data Availability Statement is currently as follows: “All relevant data are within the manuscript and in Supporting Information files.”

Please confirm at this time whether or not your submission contains all raw data required to replicate the results of your study. Authors must share the “minimal data set” for their submission. PLOS defines the minimal data set to consist of the data required to replicate all study findings reported in the article, as well as related metadata and methods (https://journals.plos.org/plosone/s/data-availability#loc-minimal-data-set-definition). For example, authors should submit the following data: - The values behind the means, standard deviations and other measures reported; - The values used to build graphs; - The points extracted from images for analysis. Authors do not need to submit their entire data set if only a portion of the data was used in the reported study. If your submission does not contain these data, please either upload them as Supporting Information files or deposit them to a stable, public repository and provide us with the relevant URLs, DOIs, or accession numbers. For a list of recommended repositories, please see https://journals.plos.org/plosone/s/recommended-repositories. If there are ethical or legal restrictions on sharing a de-identified data set, please explain them in detail (e.g., data contain potentially sensitive information, data are owned by a third-party organization, etc.) and who has imposed them (e.g., an ethics committee). Please also provide contact information for a data access committee, ethics committee, or other institutional body to which data requests may be sent. If data are owned by a third party, please indicate how others may request data access.

5. We notice that your supplementary figures are uploaded with the file type 'Figure'. Please amend the file type to 'Supporting Information'. Please ensure that each Supporting Information file has a legend listed in the manuscript after the references list.

Additional Editor Comments:

Authors are required to reply the queries raised by the reviewers. 

Reviewers' comments:

Reviewer's Responses to Questions

**Comments to the Author**

1. Is the manuscript technically sound, and do the data support the conclusions?

Reviewer #1: Yes

Reviewer #2: Yes

2. Has the statistical analysis been performed appropriately and rigorously? 

Reviewer #1: N/A

Reviewer #2: Yes

3. Have the authors made all data underlying the findings in their manuscript fully available?

Reviewer #1: Yes

Reviewer #2: Yes

4. Is the manuscript presented in an intelligible fashion and written in standard English?

Reviewer #1: Yes

Reviewer #2: Yes

5. Review Comments to the Author

Reviewer #1: The manuscript is interesting and addressesed valuable findings.

Introduction was good written and includes the aim of the study.

Materials and methods were good designed.

Results were good illustrated .

Discussion was good written.

Reviewer #2: The authors present an interesting study regarding double reading vs. AI alone vs. AI assisted interpretations of rib fractures on CT scan. The authors demonstrate that AI assisted reading of CT scans for rib fractures may help improve work flow and and potentially higher sensitivity/specificity of diagnosing rib fractures.

Further background information on the radiologists reading the scans will help prove the utility of AI. Stratifying the radiologists by their own years of practice vs. AI/AI-assisted scans would be of great interest and help improve the impact of this study.

What about pathologies associated with rib fractures? Such as PTX, hemothorax, pulmonary contusion, cardiac contusion, intercostal neurovascular bundle injury would all be of great clinical importance in addition to identifying rib fractures.

If the authors could include the specifications of the CT scanner used and the image viewing software as well would be helpful. Furthermore, the specifications of the computer system and monitors utilized by the radiologist would help elucidate these findings a bit more.

Instead of just double read CT scans, the authors should also include data from single read CT scans to see if there is an even greater need of AI assistance for radiologists operating in rural areas.

Adding the time required to perform double read vs. Ai/AI assisted reads would be very helpful to help identify any improvements in workflow and cost analysis.

6. PLOS authors have the option to publish the peer review history of their article (what does this mean?). If published, this will include your full peer review and any attached files.

Reviewer #1: No

Reviewer #2: No

---

## [Author Response · Author response to Decision Letter 0]

13 Dec 2024

Dear Editors and Reviewers,

Thank you for your thorough review and insightful comments on our manuscript. We have carefully considered each point and have provided our responses below:

Response to Reviewer #1: 

Thank you for your positive comments on our manuscript. We are gratified to hear that you found the study interesting and well-executed. Your comments encourage us and affirm the value of our research.

Response to Reviewer #2: 

Thank you for your insightful comments. We have addressed each of your points as follows:

o Background Information on Radiologists:

We appreciate your suggestion to include more background information on the radiologists involved in the study. We have expanded the relevant section in the main manuscript (Materials and Methods; 4. Image interpretation, lines 5-9) to include their years of practice, which we believe better demonstrates the utility of AI. The updated sentence now reads: " AII primary reports were conducted by 2 residents (2nd and 3rd year of residency; female, 1) and 5 attending radiologists (4 to 8 years of experience; mean, 6.6±1.7 years of experience; female, 5), with secondary readings performed by senior radiologists with over 10 years of experience (See S1 Table for details)." Additionally, we have included more comprehensive background details in the supplementary material (please see S1 Table). Additional details are provided in the supplementary material (S1 Table). While we agree that stratification by years of practice is valuable, the size constraints of our study currently limit the feasibility of conducting such an analysis. We hope to explore this issue with greater rigor in future endeavors, specifically through larger-scale, multicenter studies.

o Pathologies Associated with Rib Fractures: 

We have heeded your advice and expanded our analysis to include associated pathologies. Data on the incidence of pneumothorax, pulmonary contusions, and hemothorax have been added to the "Materials and Methods 1. Dataset collection and access" section. The updated sentence reads: "Intrathoracic injuries associated with RFs included pneumothorax (5.3%, 10 of 188), pulmonary contusions (2.1%, 4 of 188), and hemothorax (1.1%, 2 of 188)." This addition is intended to provide a more comprehensive clinical context, aligning with the importance you have highlighted. We would also like to express our gratitude for your insightful proposal. The inclusion of other pathologic changes associated with rib fractures (RFs) more fully reflects the clinical conditions of the patients in our study. As the data indicate, the incidence of pathologies associated with RFs in this study is roughly the same as the results of some of the previous studies [1], but slightly lower than others [2]. While, the distribution of fracture characteristics, as shown in S2 Fig, is generally consistent with previous researches [3,4].

o CT Scanner and Image Viewing Software and Monitor Specifications: 

We have included the specifications of the image viewing software and computer system used by the radiologists in the "Materials and Methods 1. Dataset collection and access" section. The updated sentence reads: “All images were reviewed through the picture archiving and communication system (PACS) (3.0.30500.213; Hinacom Software and Technology, Beijing, China) on a Windows 7 system (Microsoft Corporation, Redmond, Washington, United States) using professional radiographic monitors (JUSHA-M52C, JUSHA-M53AA, JUSHA-M33B and JUSHA-M33D; Nanjing Jusha Display Technology, Nanjing, China).” 

Additionally, the specifications of the CT scanners are detailed: “Scans were performed on two Siemens CT scanners (Emotion 16 and SOMATOM Definition Flash) with a 120 kV tube voltage and automated dose modulation….”

o Data from Single Read CT Scans: 

Regrettably, the collection of single read data was not feasible within the scope of our current study design. Our aim was to assess the performance of radiologists using AI assistance in a real-world clinical environment, which led us to integrate target patient images into the daily workflow through the PACS system. Given the full integration of AI-assisted diagnostic technology into our department's rib fracture diagnosis protocol, conducting single radiologist reads without AI assistance would not be practical without the radiologists' awareness, as it would deviate from our standard clinical practice. We acknowledge the importance of comparing single radiologist diagnosis with AI-assisted diagnosis, and we are aware of existing literature on this topic. Studies have consistently demonstrated that the incorporation of AI assistance in radiological diagnosis leads to a substantial enhancement in the sensitivity of single radiologists for rib fractures, concurrently preserving diagnostic specificity. The reported increase in sensitivity varies depending on the study design, ranging from 9.5% to 23.9% [5-7]. This underscores the potential clinical value of AI in enhancing diagnostic accuracy, especially in settings where radiologist resources may be limited, such as rural areas. We appreciate your guidance and will consider incorporating single read data in future prospective studies to further explore the impact of AI in various clinical scenarios, including those you've highlighted.

o Time Required for Double Read vs. AI-Assisted Reads: 

We agree that evaluating diagnostic time is a valuable research direction. Prior studies have shown that AI assistance significantly reduces diagnostic time for radiologists, with savings ranging from 73.9 seconds to 771 seconds depending on study conditions [5,7]. As a retrospective study, we were unable to directly measure diagnostic time. This limitation is acknowledged in the "Discussion" section, specifically in the penultimate paragraph where we discuss the limitations of our study. The updated content is as follows: " Third, since this is a retrospective study, the time spent on diagnosis cannot be calculated. Previous studies have consistently demonstrated the ability of AI to shorten the reading time of radiologist in diagnosing RFs in chest CT. With the aid of AI, report time savings ranged from approximately 73.9 seconds to 771 seconds, depending on the study conditions..."

Reference:

[1] Tignanelli CJ, Rix A, Napolitano LM, Hemmila MR, Ma S, Kummerfeld E. Association Between Adherence to Evidence-Based Practices for Treatment of Patients With Traumatic Rib Fractures and Mortality Rates Among US Trauma Centers. JAMA Netw Open. 2020 Mar 2;3(3):e201316. 

[2] Nummela MT, Bensch FV, Pyhältö TT, Koskinen SK. Incidence and Imaging Findings of Costal Cartilage Fractures in Patients with Blunt Chest Trauma: A Retrospective Review of 1461 Consecutive Whole-Body CT Examinations for Trauma. Radiology. 2018 Feb;286(2):696-704. doi: 10.1148/radiol.2017162429.

[3] Fukushima K, Kambe M, Aramaki Y, Ichikawa Y, Isshiki Y, Nakajima J, Sawada Y, Oshima K. Evaluation of injury threshold from the number of rib fracture for predicting pulmonary injuries in blunt chest trauma. Heliyon. 2023 Apr 6;9(4):e15278. 

[4] Barber I, Perez-Rossello JM, Wilson CR, Kleinman PK. The yield of high-detail radiographic skeletal surveys in suspected infant abuse. Pediatr Radiol 2015;45(1):69-80.

[5] Zhou QQ, Wang J, Tang W, Hu ZC, Xia ZY, Li XS, et al. Automatic Detection and Classification of Rib Fractures on Thoracic CT Using Convolutional Neural Network: Accuracy and Feasibility. Korean J Radiol. 2020;21(7):869-879. 

[6] Wu M, Chai Z, Qian G, Lin H, Wang Q, Wang L, et al. Development and Evaluation of a Deep Learning Algorithm for Rib Segmentation and Fracture Detection from Multicenter Chest CT Images. Radiol Artif Intell. 2021;3(5):e200248. 

[7] Jin L, Yang J, Kuang K, Ni B, Gao Y, Sun Y, et al. Deep-learning-assisted detection and segmentation of rib fractures from CT scans: Development and validation of FracNet. EBioMedicine. 2020;62:103106. 

We hope that these revisions adequately address your concern and strengthens the relevance of our study to clinical practice. We are committed to ensuring that our work is thorough and clinically relevant.

Thank you again for your constructive feedback.

---

## [Editor Report · Decision Letter 1]

17 Dec 2024

AI-Assisted Radiologists vs. Standard Double Reading for Rib Fracture Detection on CT Images: A Real-World Clinical Study

PONE-D-24-35102R1

Dear Dr. Xuedong Yang,

We’re pleased to inform you that your manuscript has been judged scientifically suitable for publication and will be formally accepted for publication once it meets all outstanding technical requirements.

Kind regards,

Priti Chaudhary, M.S.

Academic Editor

PLOS ONE
---

## [Editor Report · Acceptance letter]

14 Jan 2025

PONE-D-24-35102R1 

PLOS ONE

Dear Dr. Yang, 

I'm pleased to inform you that your manuscript has been deemed suitable for publication in PLOS ONE. Congratulations! Your manuscript is now being handed over to our production team.

Kind regards, 

on behalf of

Dr. Priti Chaudhary 

Academic Editor

PLOS ONE